# Association between psychological distress, lifestyle and career planning on health-related quality of life among the university students during school closure of COVID-19 pandemic in China

**Baochen Su**[1,2,3], **Zhengna Sun**[1,2,3], **Rui Chen**[1,2,3], **Hui Liu**[1,2,3], **Xixing Xu**[1,2,3], **Fanlei Kong**[1,2,3]*

**1** Department of Social Medicine and Health Management, School of Public Health, Cheeloo College of Medicine, Shandong University, Jinan, Shandong Province, China, **2** NHC Key Lab of Health Economics and Policy Research (Shandong University), Jinan, Shandong Province, China, **3** Center for Health Management and Policy Research, Shandong University (Shandong Provincial Key New Think Tank), Jinan, Shandong Province, China

* kongfanlei@sdu.edu.cn

## Abstract

The COVID-19 pandemic and associated school closures presented unprecedented challenges to university students' well-being, highlighting an urgent need to understand the factors influencing their health-related quality of life. This study aimed to explore the association between psychological distress, lifestyle, and career planning on the health-related quality of life of university students during the school closure period in China. A cross-sectional study was conducted, collecting data from 1965 Chinese college students locked down in campus during COVID-19 using a snowball sampling method via an online questionnaire platform (Wenjuan.com). Psychological distress and the health-related quality of life were measured by scales. Descriptive analysis, Chi-squared test and logistic regression analysis were employed to analyze the data. The mean physical component summary score was 47.5, while the mean mental component summary score was 36.8. Students who were women (OR=1.444, $P=0.003$), having a boy or girl friend (OR=1.379, $P=0.008$), with rural hukou (OR=1.446, $P=0.004$), with low psychological distress (OR=4.589, $P<0.001$), high physical activity intensity (OR=3.909, $P<0.001$), a regular studying schedule arrangement (OR=2.553, $P=0.008$), clear career planning (OR=1.570, $P=0.001$) during COVID-19 were more likely to report a good physical component summary. For mental component summary, lower psychological distress (OR=8.330, $P<0.001$), a regular studying schedule arrangement (OR=2.892, $P=0.001$) and keeping same job-hunting pressure (OR=1.852, $P=0.003$) were positive influencing factors, whereas having a boy or girl friend (OR=0.774, $P=0.032$) and having no clear career planning (OR=0.752, $P=0.020$) during COVID-19 were negative influencing factors.

**Data availability statement:** The dataset underlying the results cannot be shared publicly due to privacy restrictions. Data are available upon reasonable request to qualified researchers. Requests should be directed to the Science and Technology Ethics Committee of School of Public Health, Shandong University (Address: 44 Wenhuaxi Road, Cheeloo College of Medicine, Shandong University, Jinan, China. Email: lingj_li@sdu.edu.cn). The data are held securely by the institution for long-term availability per its data retention policies.

**Funding:** The author(s) declared financial support was received for the research, authorship, and/or publication of this article. This study was supported and funded by the National Natural Science Foundation of China (No. 71804094) and Fundamental Research Funds for the Central Universities (No. 2022KJGL01).

**Competing interests:** The authors have declared that no competing interests exist.

In conclusion, while health-related quality of life was generally good, lower psychological distress, healthier lifestyle habits, and clearer career planning were significantly associated with better health-related quality of life among university students during the pandemic-related school closure. These findings underscore the importance of integrating mental health support, lifestyle guidance, and career planning services into student support systems during public health crises. Future studies should develop and evaluate interventions targeting these modifiable factors.

## 1. Introduction

Since December 2019, the novel coronavirus disease 2019 (COVID-19) has spread from Wuhan city to different areas in China and around the world [1]. As of December 2022, there are 9311491 Chinese people had been infected with COVID-19 [2]. This pandemic has disrupted our normal pace of life, and further brought many health problems [3]. No reliable COVID-19-specific therapeutics were available at the time [4], leading almost all universities in China to adopt closure policies to ensure the health of university students. School closure is an important way for many Chinese universities in response to the COVID-19. During the semester, all students' studies and lives were on campus, and they can't enter or leave the campus unless they need medical treatment. Many gathering activities were canceled, and many courses were changed to online learning in the dormitories [5]. This closed and repressive environment and the reduction of collective activities will bring some mental and physical problems [6].

Campus lockdowns abruptly removed the structured social, physical and academic environment that normally sustains student health [7]. While school closures during the COVID-19 pandemic reduced infection risks, they simultaneously triggered a cascade of challenges for university students including academic procrastination, deteriorating mental and physical health, diminished social support, heightened loneliness, and altered Health-Related Quality of Life (HRQOL) [8–10] And school closure may make the situation worse [11]. However, most evidence is based on convenience samples collected during the first semester of the pandemic; little is known about the sustained impact of strict "closed-campus" policies—unique to mainland China—where students were physically confined to dormitories for entire academic years. Consequently, the health status, the quality of life of the university students and its influencing factors during the school closure of the COVID-19 was worth further research.

Health-related quality of life (HRQOL) is defined as an individual's satisfaction or happiness with the quality of their physical and psychological distress and is deemed a comprehensive assessment tool of a population's health status [12]. Previous research found that adolescents with internalizing and externalizing psychological distress problems had the lowest HRQOL [13]. University students with good healthy lifestyles may have better HRQOL [14]. While those who had a boy or girl friend, female and physical activity tended to a positive effect on the HRQOL [15]. However

the COVID-19 pandemic made these relationships different and more complex [16]. COVID-19 as a public health emergency has been proved by researches that it may affect many aspects of university students, such as mental health [17,18] and physical health problems [19].

Demographic factors have always been important influencing factors on HRQOL, and even some studies focus on it as an independent and important influencing factor [20]. Previous study has shown that gender differences have significant impact on the quality of life of medical students [21]. With respect to the medical and psychological aspects, one of the most eminent and influential Persian philosophers and scientists Avicenna held that lovesickness brings disease to the soul and body of a lover [22]. In addition, many basic scientific studies have shown a very close relationship between love and health [23]. So it will be very interesting to explore the association between love and the HRQOL of university students in the special situation where many couples cannot see each other for a long time during the school closure of COVID-19. A Chinese study found that university students who have the urban hukou (registration) enjoyed higher subjective social status, which had a clear protective effect against anxiety and depression symptoms [24]. And another study from China showed that the non-only children were more likely to develop the symptoms of anxiety and depression than only children, during the outbreak of COVID-19 in China [25].

Lifestyle habits has been proven to be one of the most important issues affecting the physical and mental health of the students [26]. A German study found that University students who had better sleep tend to had better HRQOL [27], while the adolescents with regular sports activity would had higher HRQOL during stay at home than those with non-regular sports activity [28]. Previous research on lifestyle habits has focused on smoking and drinking which are relatively rare among the university students while the physical activity [29] and sleep habits [30] are very common among the university students which changed a lot during the lockdown period and have a significant impact on their lives.

Literature reviews showed there was little research on the relationship between career planning and HRQOL among the university students currently [31]. A previous study found that there was a significant difference on the student career perception before and during the COVID-19 [32], while the changes in the student career perception will bring the changes in their career planning and even career pressure [33,34]. Thus, the school closure of the COVID-19 may affect the students' career planning.

To conclude, some researches focused on the groups of university students during COVID-19 [35,36]. Few researches focused on gender differences on the quality of life during the COVID-19 pandemic [37]. Only one study had explored the relationship between career planning and HRQOL during the COVID-19 pandemic [32]. Let alone examined the association between psychological distress, lifestyle, and career planning on HRQOL among the university students especially during the school closure of COVID-19. Thus, this study aimed to clarify the association between psychological distress, lifestyle and career planning on HRQOL of the university students during the school closure of COVID-19.

## 2. Materials and methods

### 2.1. Data and sample

Cross-sectional survey design and anonymous online questionnaire were used to assess the quality of life of university students during the school closure of COVID-19. The survey was conducted through an online platform called "Wenjuan.com" (commonly used in China), using a snowball sampling method. Initial respondents were recruited via university student WeChat groups; they were then encouraged to share the questionnaire link with their peers who also met the eligibility criteria, thereby expanding the sample through chain-referral. Data collection took place from May 5th to May 12th, 2022, during the period of school closure. All participants provided informed consent electronically by agreeing to participate before starting the questionnaire. All participants have provided informed consent through answering questions online.

The snowball sampling method was used due to the difficulty of accessing the nationally dispersed university student population through traditional random sampling during strict campus lockdowns. This approach effectively reached the target population through participants' social networks in the absence of a reliable sampling frame.

Participants were recruited according to the following protocol-defined eligibility requirements: only Chinese students who were officially registered at a mainland university and were obliged to reside on campus under the institution's COVID-19 "closed-campus" policy were eligible; international students, those living off-campus, and any individuals with a pre-existing major physical or psychiatric disorder that might independently affect the quality of life were excluded.

The minimum sample size was calculated using the cross-sectional formula $n = [Z^2_{1-\alpha/2} \times P \times (1-P) / d^2]$, with $Z = 1.96$ (95% CI), $P = 0.25$, $d = 0.05$ and a design effect of 1.3, yielding 416 participants after 10% invalid allowance. To ensure statistical power and account for invalid responses, after excluding incomplete responses and malicious filling, a total of 1965 questionnaires were used in this study well exceeding the minimum requirement.

## 2.2. Measurements

### 2.2.1. Health-Related Quality of Life (HRQOL).

The Short-Form Health Survey (SF-12), consisting of 12 items drawn from the eight subscales of the SF-36 [38], is widely used to measure HRQOL [39] and has been tested as a scale with good reliability and validity [40]. The Chinese-translated version of the SF-12 questionnaire has been reported to be suitable for the adolescents in China [41]. In this study, HRQOL of University students during the school closure of COVID-19 was assessed by SF-12 scores that were divided into the mental component summary (MCS) and the physical component summary (PCS) scores for calculation [42]. Each participant's MCS and PCS were dichotomized by the cutoff point of the first quartile of MCS and PCS scores, and we defined poor HRQOL as scores lower than that of the first quartile [43,44]. In this study the Cronbach's alpha of this scale was 0.83.

### 2.2.2. Psychological distress.

Psychological distress in this study was measured through The Kessler Psychological disease scale(K10) The K10 scale consists of 10 items that measure the frequency of occurrence of non-specific psychological symptoms related to anxiety and stress levels experienced in the past four weeks. Each item is rated on a scale of five levels: all the time, most of the time, some of the time, occasionally, and almost never. Scores are assigned to each level as follows: 5 points for all the time, 4 points for most of the time, 3 points for some of the time, 2 points for occasionally, and 1 point for almost never. The scores of the 10 items are then summed, and individuals' mental health status is classified into four levels based on the total score of the K10: 10–19 points (Level 1, low psychological distress), 20–24 points (Level 2, relatively low psychological distress), 25–29 points (Level 3, relatively high psychological distress), and 30–50 points (Level 4, high risk psychological distress). K10 has been tested as a scale with good reliability and validity [36,45,46]. In this study the Cronbach's alpha of this scale was 0.93.

### 2.2.3. Lifestyle.

Lifestyle was mainly measured through the dimensions of studying schedule arrangement and physical activity intensity. studying schedule arrangement were measured by two questions "What is your studying schedule arrangement during the school closure of the COVID-19?" and "Has your sleep condition changed before and after the school closure?" Physical activity intensity was calculated by the total score of three questions "How was the intensity of your physical activity during the past month?", "How many minutes did you spend on the physical activity?", "How many times have you engaged in the physical activity within a month?" Each question was divided into five levels: 1–5. The total score was calculated as intensity × (time − 1) × frequency, with a maximum score of 100 and a minimum score of 0. The assessment criteria were as follows: low physical activity level is ≤ 19 points, moderate physical activity level is 20–42 points, and high physical activity level is ≥ 43 points [47,48]. In this study the Cronbach's alpha of this scale was 0.86.

### 2.2.4. Career planning.

Career planning was captured by four questions: "Do you have clear future planning", "Does your future planning change because of the school closure of the COVID-19", "How about your current job-hunting pressure" and "Changes in your job-hunting pressure under the epidemic of the school closure of the COVID-19". Each question was analyzed as an independent variable in this study.

## 2.3. Statistical analysis

All statistical analyses were performed using SPSS (Statistical Package for the Social Sciences, IBM Corp. Released 2016. IBM SPSS Statistics for Windows, Version 25.0. IBM Corp., Armonk, NY, USA), and $p$-values less than 0.05 were regarded as statistically significant. Chi-square test was used to calculate the differences in MCS and PCS among the subgroups of each categorical variable. After univariate analyses, statistically significant variables were included in the logistic regression analyses. Three binary logistic regression models with an enter method were adopted to explore the associations between independent variables and HRQOL. Meanwhile, crude odds ratios (OR) and 95% confidence intervals (95% CI) were calculated. First, social demographic characteristics and psychological distress entered Model 1, then the indicators of lifestyle habits were brought into Model 2, and finally the variables of future planning were added to Model 3.

## 2.4. Ethics statement

The study was designed in accordance with the tenets of the Declaration of Helsinki, 1996 and were approved by Shandong University Institutional Ethics Committee. (Task no. LL20220425) The studies were conducted in accordance with the local legislation and institutional requirements. Written informed consent for participation in this study was provided by the participants' legal guardians/next of kin. Written informed consent was obtained from the individual(s) for the publication of any potentially identifiable images or data included in this article. Human Ethics and Consent to Participate declarations: not applicable.

## 3. Results

### 3.1. Basic characteristics of the participants

The basic information about the participants was provided in Table 1. All students were Chinese studied in Chinese universities with an average age of 22.25±1.71. It was shown that 59.8% of the sample was female. 55.8% of participants were rural registered residence. As for the region of the participants, 74.9% of participants were recruited from universities in northern China and 25.1% from southern China. Chi-square tests indicated no significant between-region differences for either PCS or MCS ($P=0.679$ and 0.134)

### 3.2. PSC and MCS of HRQOL

As shown in Table 1,2, factors that were statistically significant different in PCS included gender ($P<0.001$), only child ($P<0.05$), hukou ($P<0.05$),in love($P<0.001$), psychological distress ($P<0.001$) studying schedule arrangement ($P<0.001$）physical activity intensity ($P<0.001$) clear career planning($P<0.001$) the change of career planning during COVID-19 ($P<0.001$) job-hunting pressure ($P<0.001$). The Changes of job-hunting pressure during the school closure of the COVID-19 ($p<0.05$)

As for MCS, there were statistically significant differences in love ($p<0.05$) psychological distress ($p<0.001$) studying schedule arrangement($p<0.001$）clear career planning ($p<0.001$) job-hunting pressure ($p<0.001$) Changes of job-hunting pressure during the school closure of the COVID-19($p<0.001$).

### 3.3. Relationship between the related variables and PCS

Table 3 display the $p$-values, OR, and 95% CI for the logistic analysis on association between the related variables and PCS. The collinearity diagnostics' results revealed that the tolerances of all independent variables were higher than 0.1, and the variance inflation factors were far less than 10, suggesting that there was no multicollinearity between independent variables in three logistic regression models.

In Table 3, the results of Model 1 indicated that gender, hukou, being loved with others, and psychological distress were statistically significant factors of PCS. When variables of lifestyle habits entered Model 2, these four variables were

**Table 1. Basic characteristics of the participants.**

| Variables | Total | Good PCS | Poor PCS | *P* | Good MCS | Poor MCS | *P* |
|---|---|---|---|---|---|---|---|
|  | n (%) | n (%) | n (%) |  | n (%) | n (%) |  |
| **Age** | 22.25±1.71 | 23.0±1.41 | 21.5±2.12 | 0.801 | 21.8±2.40 | 21.6±2.45 | 0.081 |
| **Gender** |  |  |  | <0.001 |  |  | 0.316 |
| Male | 790(40.2) | 555(70.3) | 235(29.7) |  | 606(76.7) | 184(23.3) |  |
| Female | 1175(59.8) | 943(80.3) | 232(19.7) |  | 898(75.1) | 297(24.9) |  |
| **hukou** |  |  |  | 0.022 |  |  | 0.305 |
| Rural | 1096(55.8) | 814(74.3) | 282(25.7) |  | 818(74.6) | 278(25.4) |  |
| City | 869(44.2) | 684(78.7) | 185(21.3) |  | 666(76.6) | 203(23.4) |  |
| **Region** |  |  |  | 0.679 |  |  | 0.134 |
| northern China | 1472(74,9) | 1119(74.7) | 353(75.6) |  | 1100(74.1) | 372(77.3) |  |
| southern China | 493(25.1) | 379(25.3) | 114(24.4) |  | 384(25.9) | 109(22.7) |  |

Notes: HRQOL=health-related quality of life, MCS=mental component summary, PCS=physical component summary.

still statistically significant. Studying schedule arrangement regularly most of the time emerged as statistically significant. Model 3 presented that psychological distress. Lifestyle habits and career planning were statistically associated with PCS. Specifically, university students during COVID-19 pandemic in China who were female (*P*=0.003, OR=1.444), from rural area(*P*=0.004,OR=1.446),were not in love(*P*=0.008, OR=1.379), lower possibility of mental disease(*P*<0.001, OR=3.909), could studying schedule arrangement regular most of the time (*P*=0.003, OR = 2.553),could had a high physical activity intensity(*P*<0.001, OR = 4.589), had a clear career planning(*P*=0.001,OR=1.570), had no changes of life planning during the school closure of the COVID-19 (*P*<0.001, OR=1.988) were more likely to have good PCS.

### 3.4. Relationship between the related Variables and MCS

As shown in Table 4, the university students during the school closure of the COVID-19 who had lower possibility of disease (*P*<0.001, OR=8.330), studying schedule arrangement quite regular (*P*=0.001, OR=2.892) and whose job-hunting pressure not change (*P*=0.003, OR=1.852) were more likely to had good MCS while for those who were not in love (*P*=0.032, OR=0.774), had no clear career planning (*P*=0.020, OR=0.752), the scores were reversed.

## 4. Discussion

### 4.1. Association between Social Demographic Characteristics and HRQOL

In terms of demographic characteristics, the significant predictors of HRQOL were gender, love and hukou. Some previous studies showed that females scored significantly lower than males for HRQOL [49]. But in our research male was a risk factor for PCS of SCSCC. There are significant differences in the ways that males and females participate in activities. Males often gravitate towards physical activities and team-based projects during their free time, whereas females tend to prioritize social engagements and personal interests [50]. On average, males tend to engage in more physical activity and for longer periods of time than females [51]. The closing of sports facilities and prohibition of group gatherings during school closures and pandemic prevention measures disproportionately impacted the physical activities of males as compared to females. As a result, males experienced a significant decline in PCS, which has been more pronounced than that experienced by females. The reduction in physical activity due to external factors has led many males to increase usage of electronic devices, often with a higher frequency of use. As a result, the decline in physical health has become apparent, indicating the negative impact of their physical health [52].

**Table 2. Descriptive for HRQOL in the mental and physical dimensions of the participants.**

| Variables | Total | Good PCS | Poor PCS | P | Good MCS | Poor MCS | P |
|---|---|---|---|---|---|---|---|
| | n (%) | n (%) | n (%) | | n (%) | n (%) | |
| **One child only** | | | | 0.039 | | | 0.819 |
| Yes | 845(43.0) | 625(74.0) | 220(26.0) | | 636(75.3) | 209(24.7) | |
| No | 1120(57.0) | 873(77.9) | 247(22.1) | | 848(75.7) | 272(24.3) | |
| **Love** | | | | <0.001 | | | 0.015 |
| Be in love | 899(45.8) | 644(71.6) | 255(28.4) | | 702(78.1) | 197(21.9) | |
| Not in love | 1066(54.2) | 854(80.1) | 212(19.9) | | 782(73.4) | 284(26.6) | |
| **Psychological distress** | | | | <0.001 | | | <0.001 |
| Lower psychological distress | 948(48.2) | 823(86.8) | 125(13.2) | | 851(89.8) | 97(10.2) | |
| Low psychological distress | 470(23.9) | 361(76.8) | 109(23.2) | | 345(73.4) | 125(26.6) | |
| High psychological distress | 289(14.7) | 166(57.4) | 123(42.6) | | 174(60.2) | 115(39.8) | |
| Higher psychological distress | 258(13.1) | 148(57.4) | 110(42.6) | | 114(44.2) | 144(55.8) | |
| **Studying schedule arrangement** | | | | <0.001 | | | <0.001 |
| Quiet Regular | 378(19.2) | 301(79.6) | 77(20.4) | | 324(85.7) | 54(14.3) | |
| Regular most of the time | 1067(54.3) | 851(79.8) | 216(20.2) | | 848(79.5) | 219(20.5) | |
| Not very regular | 457(23.3) | 309(67.6) | 148(32.4) | | 286(62.6) | 171(37.4) | |
| Not regular | 63(3.2) | 37(58.7) | 26(41.3) | | 26(41.3) | 37(58.7) | |
| **Physical activity intensity** | | | | <0.001 | | | 0.234 |
| Small amount | 1411(71.8) | 1011(71.7) | 400(28.3) | | 1051(74.5) | 360(25.5) | |
| Medium amount | 366(18.6) | 316(86.3) | 50(13.7) | | 286(78.1) | 80(21.9) | |
| Large amount | 188(9.6) | 171(91.0) | 17(9.0) | | 147(78.2) | 41(21.8) | |
| **Clear career planning** | | | | 0.001 | | | <0.001 |
| Yes | 1270(64.6) | 937(73.8) | 333(26.2) | | 1002(78.9) | 268(21.1) | |
| No | 695(35.4) | 561(80.7) | 134(19.3) | | 482(69.4) | 213(30.6) | |
| **The change of career planning during COVID-19** | | | | <0.001 | | | 0.249 |
| Yes | 753(38.3) | 487(64.7) | 266(35.3) | | 558(74.1) | 195(25.9) | |
| No | 1212(61.7) | 1011(83.4) | 201(16.6) | | 926(76.4) | 286(23.6) | |
| **Job-hunting pressure** | | | | <0.001 | | | <0.001 |
| Not at all | 135(6.9) | 105(77.8) | 30(22.2) | | 110(81.5) | 25(18.5) | |
| A little | 361(18.4) | 255(70.6) | 106(29.4) | | 289(80.1) | 72(19.9) | |
| Some | 1097(55.8) | 880(80.2) | 217(19.8) | | 858(78.2) | 239(21.8) | |
| Quite a lot | 309(15.7) | 218(70.6) | 91(29.4) | | 195(63.1) | 114(36.9) | |
| Huge | 63(3.2) | 40(63.5) | 23(36.5) | | 32(50.8) | 31(49.2) | |
| **The Changes of job-hunting pressure during the school closure of the COVID-19** | | | | 0.014 | | | <0.001 |
| Become much lower | 33(1.7) | 28(84.8) | 5(15.2) | | 29(87.9) | 4(12.1) | |
| Become lower | 112(5.7) | 77(68.8) | 35(31.3) | | 82(73.2) | 30(26.8) | |
| No change | 668(34.0) | 531(79.5) | 137(20.5) | | 545(81.6) | 123(18.4) | |
| Become higher | 915(46.6) | 694(75.8) | 221(24.2) | | 679(74.2) | 236(25.8) | |
| Become much higher | 237(12.1) | 168(70.9) | 69(29.1) | | 149(62.9) | 88(37.1) | |

Notes: HRQOL = health-related quality of life, MCS = mental component summary, PCS = physical component summary.

**Table 3. Binary logistic regression for relationships between related variables and PCS.**

| Variables | Model1 | | Model2 | | Model3 | |
|---|---|---|---|---|---|---|
| | OR (95%CI) | *P* | OR (95%CI) | *P* | OR (95%CI) | *P* |
| **Gender** | | | | | | |
| Male | 1 | | 1 | | 1 | |
| Female | 1.521(1.213-1.906) | <0.001 | 1.664(1.318-2.102) | <0.001 | 1.444(1.135-1.838) | 0.003 |
| **One child only** | | | | | | |
| Yes | 1 | | 1 | | 1 | |
| No | 1.222(0.957-1.559) | 0.108 | 1.239(0.966-1.591) | 0.092 | 1.186(0.920-1.528) | 0.189 |
| **Hukou** | | | | | | |
| Rural | 1 | | 1 | | 1 | |
| City | 1.450(1.138-1.847) | 0.003 | 1.430(1.116-1.831) | 0.005 | 1.446(1.125-1.859) | 0.004 |
| **Love** | | | | | | |
| Be in love | 1 | | 1 | | 1 | |
| Not in love | 1.536(1.231-1.906) | <0.001 | 1.651(1.313-2.077) | <0.001 | 1.379(1.085-1.752) | 0.008 |
| **Psychological distress** | | | | | | |
| Higher psychological distress | 1 | | 1 | | 1 | |
| High psychological distress | 0.974(0.690-1.376) | 0.881 | 0.866(0.602-1.245) | 0.436 | 0.864(0.593-1.259) | 0.447 |
| Low psychological distress | 2.371(1.700-3.306) | <0.001 | 2.089(1.472-2.963) | <0.001 | 2.004(1.395-2.879) | <0.001 |
| Lower psychological distress | 4.783(3.490-6.554) | <0.001 | 4.217(2.995-5.937) | <0.001 | 3.909(2.722-5.614) | <0.001 |
| **Studying schedule arrangement** | | | | | | |
| Not Regular | | | 1 | | 1 | |
| Not very regular | | | 1.437(0.794-2.601) | 0.231 | 1.464(0.796-2.690) | 0.22 |
| Regular most of the time | | | 2.324(1.292-4.179) | 0.005 | 2.553(1.388-4.697) | 0.003 |
| Quite regular | | | 1.903(1.007-3.597) | 0.048 | 2.266(1.171-4.384) | 0.015 |
| **Physical activity intensity** | | | | | | |
| Small | | | 1 | | 1 | |
| Middle | | | 2.805(2.000-3.935) | <0.001 | 2.843(2.015-4.010) | <0.001 |
| Big | | | 4.802(2.807-8.213) | <0.001 | 4.589(2.662-7.909) | <0.001 |
| **Clear career planning** | | | | | | |
| Yes | | | | | 1 | |
| No | | | | | 1.57(1.207-2.041) | 0.001 |
| **The change of career planning during COVID-19** | | | | | | |
| Yes | | | | | 1 | |
| No | | | | | 1.988(1.557-2.539) | <0.001 |
| **Job-hunting pressure** | | | | | | |
| Not at all | | | | | 1 | |
| A little but it can be ignored | | | | | 1.275(0.715-2.276) | 0.41 |
| Some but adjustable | | | | | 1.136(0.663-1.945) | 0.642 |
| More and affect life | | | | | 1.115(0.624-1.990) | 0.714 |
| Huge | | | | | 1.135(0.525-2.451) | 0.748 |
| **Changes in job-hunting pressure during COVID-19** | | | | | | |
| Become very big | | | | | 1 | |
| Become big | | | | | 1.003(0.686-1.466) | 0.986 |
| No change | | | | | 0.887(0.585-1.343) | 0.571 |
| Become small | | | | | 0.637(0.361-1.123) | 0.119 |
| Become very small | | | | | 1.001(0.335-2.989) | 0.998 |

Notes:.M1 = Demography + psychological distress, M2 = M1 + lifestyle habits, M3 = M2 + career planning MCS = mental component summary, PCS = physical component summary, OR = crude odds ratios, 95% CI = 95% confidence intervals

**Table 4. Binary logistic regression for relationships between related variables and MCS.**

| Variables | Model 1 | | Model 2 | | Model 3 | |
|---|---|---|---|---|---|---|
| | OR (95%CI) | *P* | OR (95%CI) | *P* | OR (95%CI) | *P* |
| **Love** | | | | | | |
| Be in love | 1 | | 1 | | 1 | |
| Not in love | 0.717(0.572–0.899) | 0.004 | 0.765(0.609-0.962) | 0.022 | 0.774(0.613-0.978) | 0.032 |
| **Psychological distress** | | | | | | |
| Higher possibility of disease | 1 | | 1 | | 1 | |
| High possibility of disease | 1.928(1.370-2.714) | <0.001 | 1.645(1.158-2.336) | 0.005 | 1.620(1.134-2.313) | 0.008 |
| Low possibility of disease | 3.559(2.581-4.908) | <0.001 | 2.984(2.144-4.152) | <0.001 | 2.945(2.104-4.121) | <0.001 |
| Lower possibility of disease | 11.313(8.176-15.655) | <0.001 | 8.765(6.243-12.306) | <0.001 | 8.330(5.866-11.829) | <0.001 |
| **Studying schedule arrangement** | | | | | | |
| Not Regular | | | 1 | | 1 | |
| Not very regular | | | 1.644(0.920-2.938) | 0.093 | 1.612(0.796-2.690) | 0.113 |
| Regular most of the time | | | 2.883(1.630-5.101) | <0.001 | 2.669(1.486-4.791) | 0.001 |
| Quite regular | | | 3.166(1.683-5.958) | <0.001 | 2.892(1.514-5.524) | 0.001 |
| **Whether had a clear career plan** | | | | | | |
| Yes | | | | | 1 | |
| No | | | | | 0.752(0.592-0.955) | 0.020 |
| **Job-hunting pressure** | | | | | | |
| Not at all | | | | | 1 | |
| A little | | | | | 1.217(0.683-2.169) | 0.504 |
| Some | | | | | 1.060(0.622-1.808) | 0.830 |
| Quite a lot | | | | | 1.078(0.605-1.919) | 0.799 |
| Huge | | | | | 1.110(0.517-2.384) | 0.788 |
| **Changes in job-hunting pressure** | | | | | | |
| Become very big | | | | | 1 | |
| Become big | | | | | 1.392(0.966-2.006) | 0.076 |
| No change | | | | | 1.852(1.240-2.766) | 0.003 |
| Become small | | | | | 1.113(0.630-1.967) | 0.713 |
| Become very small | | | | | 1.654(0.516-5.307) | 0.397 |

Notes: M1 = Demography + psychological distress, M2 = M1 + lifestyle habits, M3 = M2 + career planning, MCS = mental component summary, PCS = physical component summary, OR = crude odds ratios, 95% CI = 95% confidence intervals.

Having a partner was a risk factor for PCS of HRQOL, but a protective factor for MCS of HRQOL. In the context of the COVID lockdowns, it is inevitable that there will be interruptions to the daily lives of individuals and their partners. In addition, the lack of physical activity and reduced physical contact can also contribute to a decrease in PCS [53]. Research indicates that individuals who were satisfied with the quality of their romantic relationships score higher than those with poor relationship quality or no relationship at all [54]. This suggests that higher quality interpersonal relationships can be a protective factor against mental health problems during the COVID-19 pandemic [55]. Therefore, having a partner during the COVID-19 pandemic and school closure can often have a positive impact on MCS.

Previous literature research has shown that individuals with urban household registration tend to have more social resources and connections, as well as better mental health status than those with rural household registration [56,57]. And this increased social support and better mental health status have a positive impact on HRQOL especially in COVID-19 pandemic and school closure [58].

## 4.2. Association between psychological distress and HRQOL

In this study, Psychological distress has been found to be associated with HRQOL, which is consistent with other research, indicating that the COVID-19 pandemic and isolation measures can lead to an increase in psychological distress [59,60]. Previous research found that Chinese adults experienced stress, helplessness, and lower quality of life due to the COVID-19 pandemic [61]. Compared with Chinese COVID-19 patients without depression, patients with depression reported lower levels of quality of life [62]. The COVID-19 pandemic has resulted in an increase in psychological distress and a significant impact on quality of life by affecting the social ecosystem [63]. The implementation of lockdown measures exacerbates these issues, as demonstrated in a study conducted in Italy and Israel [64]. Those with higher levels of anxiety and psychological distress tend to have lower quality of life [59,60,62].

## 4.3. Association between lifestyle and HRQOL

This study found that lifestyle is an important factor affecting HRQOL. In our study, students who had a regular study schedule often had better HRQOL. And physical activity intensity mainly affects students' PCS. Students who had big physical activity intensity often had higher scores on PCS. A study of Chinese university students also showed a positive relationship between exercise intensity and quality of life [65]. This finding was basically consistent with the results of this study. But previous studies have also shown that exercise had a greater impact on MCS in university students [66]. The results of this study indicated that exercise had a greater impact on PCS during school closures due to the COVID-19 pandemic. This may be due to the significant decrease in activities caused by the school closure due to the epidemic, and even difficulty in leaving the dormitory. Many studies have shown that reducing physical activity may cause serious harm to the health of college students [67,68]. Thus, during the lockdown period, the relationship between exercise intensity and PCS may become more prominent.

Research has shown that a regular studying schedule arrangement can affect physical health in various aspects of physiology [69,70]. Previous studies have shown that regular studying schedule arrangement is a particularly positive factor in improving quality of life. This was basically consistent with the conclusion of our study [67,68,70–73].

## 4.4. Association between career planning and HRQOL

There were relatively few previous studies on the association between career planning and HRQOL for university students, not to mention during the COVID-19 pandemic and school closure. Thus, it was difficult to make comparisons between our research and existing research. Our study showed that students who had clear career planning and experienced a big change in job-hunting pressure had higher scores of MCS. Previous literature has shown that the changes in the job-hunting situation caused by the COVID-19 pandemic and school closure have triggered a lot of anxiety and depression in many university students [74,75]. University students who were under greater job-hunting pressure due to the impact of the COVID-19 pandemic and school closure often experience more mental health problems and increased anxiety and depression, resulting in a decline in their HRQOL [76,77]. The above research results may explain the association between career planning and MCS in our research.

This study found students who had no clear career planning and those who had no changes of career planning during the COVID-19 pandemic had higher scores on PCS. Previous literature indicated that the pandemic has brought unprecedented pressure and enormous disruptions in daily lives [78–80]. Before and after the pandemic, students' job-hunting intentions, attitudes towards job-hunting, and career choices had undergone significant changes [80]. Thus, those who had a clear career planning and those had no changes in career planning during the COVID-19 pandemic and school closure were less likely to have some mental or physical health problems due to the worry about their careers and futures caused by COVID-19 pandemic and school closure.

 

## 4.5. Implications

To improve the HRQOL of university students during school closure due to acute infectious diseases, the government may take the following measures. First, more attention should be paid to the HRQOL of female students and the students whose hukou in countryside. Schools can provide them with more social support, such as holding online parties or some interesting activities. For people in love, appropriate psychological guidance can be given, and long-distance students can be encouraged to actively carry out outdoor activities. Second, holding some sports competitions and interesting sports activities is a good way to improve PCS during school closure. Turn off the lights and power off regularly at night and set a fixed catering time. Conducting more education and counseling on psychological distress and setting up psychological counseling clinics during the school closure period. Meanwhile, students should be educated and guided in life planning to help them have a stable life planning which is not affected by changes in social situation at ordinary times. Treating large-scale public health emergencies rationally, setting up correct views of occupation choice. University students are a vulnerable group in the event of sudden public health emergencies; thus, we should pay more attention to the various problems faced by university students in this special situation. When new infectious diseases and other health emergencies arise, the health status of college students deserves attention, and intervention measures tailored to their characteristics should be implemented in advance. Future research should explore various health indicators of university students in this scenario and explore the mechanisms of their impacts. In addition to psychology, life, and occupation, other variables could be expanded on this basis for more comprehensive exploration and research. Some appropriate interventions should be studied and applied.

## 4.6. Limitations

There are several limitations to this study. First, due to the COVID-19 pandemic and the strict closure of schools in China, many questionnaire surveys could not be conducted normally. Thus, our study could only use a cross-sectional online questionnaire survey with non-random sampling, which may result in some bias in the representativeness of the sample. Second, due to the lack of a systematic scale for lifestyle habits and career planning in the questionnaire, we only selected some relevant indicators to conduct our evaluation, which was expected to be remedied in the follow-up study. Finally, we used data from a cross-sectional study, so the causal relationship could not be predicted.

## 5. Conclusion

Lower psychological distress, better lifestyle and clearer career planning were significantly associated with the better HRQOL of university students during the school closure of the COVID-19. Maintaining a regular studying schedule arrangement and a relatively high-intensity of exercise during the school closure of the COVID-19 may have a good impact on maintaining a good HRQOL of university students. Intervention measures derived from this research could be put into practice for governments and universities in the future public health emergencies.

## Acknowledgments

The research team greatly appreciates the support and the research participants for their cooperation and support. The authors are grateful that Insha Yousuf Ellahie and Tawhida Mwinyi Tallib conducted the English language proofreading of the revised manuscript; Jiajia Li and Shixue Li gave many valuable suggestions on the revision of manuscript during the peer review procedure.

## Author contributions

**Conceptualization:** Baochen Su, Fanlei Kong.

**Data curation:** Baochen Su, Rui Chen, Hui Liu, Xixing Xu, Fanlei Kong.

**Formal analysis:** Baochen Su, Rui Chen, Hui Liu.

**Investigation:** Baochen Su, Rui Chen.

**Methodology:** Baochen Su, Rui Chen, Fanlei Kong.

**Resources:** Baochen Su.

**Software:** Baochen Su, Rui Chen.

**Supervision:** Baochen Su, Fanlei Kong.

**Validation:** Baochen Su, Fanlei Kong, Zhengna Sun.

**Visualization:** Baochen Su, Rui Chen, Fanlei Kong, Zhengna Sun.

**Writing – original draft:** Baochen Su, Fanlei Kong.

**Writing – review & editing:** Baochen Su, Fanlei Kong, Zhengna Sun.

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
