## [Decision Letter · Decision Letter 0]

22 Dec 2025

Dear Dr. Kong,

Thank you for submitting your manuscript to PLOS ONE. After careful consideration, we feel that it has merit but does not fully meet PLOS ONE’s publication criteria as it currently stands. Therefore, we invite you to submit a revised version of the manuscript that addresses the points raised during the review process.

**ACADEMIC EDITOR:**

Dear Authors,

Thank you for your efforts and for choosing Plos One to publish your investigation.

Three reviewers have evaluated your manuscript and minor revisions are suggested.

Please, complete all the queries and send them back at your earliest convenience.

Best regards.

We look forward to receiving your revised manuscript.

Kind regards,

Javier Fagundo-Rivera, PhD

Academic Editor

PLOS One

Journal Requirements:

“The author(s) declared financial support was received for the research, authorship, and/or publication of this article. This study was supported and funded by the National Natural Science Foundation of China (No. 71804094) and Fundamental Research Funds for the Central Universities (No. 2022KJGL01).”

**Additional Editor Comments:**

Dear Authors,

Thank you for your efforts and for choosing Plos One to publish your investigation.

Three reviewers have evaluated your manuscript and minor revisions are suggested.

Please, complete all the queries and send them back at your earliest convenience.

Best regards.

Reviewers' comments:

Reviewer's Responses to Questions

**Comments to the Author**

1. Is the manuscript technically sound, and do the data support the conclusions?

Reviewer #1: Yes

Reviewer #2: Yes

Reviewer #3: Yes

2. Has the statistical analysis been performed appropriately and rigorously?

Reviewer #1: Yes

Reviewer #2: Yes

Reviewer #3: Yes

3. Have the authors made all data underlying the findings in their manuscript fully available?

Reviewer #1: No

Reviewer #2: No

Reviewer #3: Yes

4. Is the manuscript presented in an intelligible fashion and written in standard English?

Reviewer #1: Yes

Reviewer #2: Yes

Reviewer #3: No

**Reviewer #1: SEE DOCUMENT ATTACHED.**

The manuscript “Association Between Psychological Distress, Lifestyle and Career Planning on Health-Related Quality of Life Among the University Students during the COVID-19 Pandemic School Closure in China,” presents original cross-sectional design research examining the determinants of Health-Related Quality of Life (HRQOL) in 1965 Chinese university students during the COVID-19 school closure. I now submit some minor revisions I recommend the authors include, based on PLOS ONE’s criteria.

**Reviewer #2:**

I commend the authors for this interesting manuscript. Most university students went through a lot during the COVID-19 pandemic, resulting in the closure of schools.

I believe the manuscript will be strengthened after considering these few points.

1. Under the Materials and Methods, there should be exclusion and inclusion criteria to understand the category of group that the study considered. it is also reported that the snowballing sampling method was used in collecting the online data. It should be clearly stated in the manuscript how it was done during the data collection.

2. Under results, participants' characteristics are incomplete because we are sure of the age group of students you considered in the study and whether they are all nationals or not.

**Reviewer #3:**

Thank you for the opportunity to review this paper. The idea of the paper is good, but more work needs to be done on it. My suggestions are as follows:

1. The abstract should present a clear overview of the study's objectives and the general findings. However, the first paragraph should start with a sentence about the importance of the study and then the purpose of the research.

2. In the methodology part of the abstract, it is not mentioned about the population and the data gathering tool. Please complete it.

3. In the result part of the abstract, the authors should bring the complete words instead of the acronyms they have used (MCS = mental component summary, PCS = physical component summary)

4. Some claims and sentences do not have references. For instance, lines 119-121-47. Please check the article.

5- The "Introduction" should be developed, modified, rearranged, and divided into some paragraphs based on scientific writing. To strengthen the paper, I would suggest expanding the introduction and bringing more from the literature about the idea of this paper; that would make a stronger contribution to the international audience.

6- The authors should justify why they have used snowball sampling in this study, and mention the population size and how they calculated the sample size in the methodology.

7. For better readability, the authors should divide the very long table into some specific and concise tables in the article.

8. There are some mistakes, so the language and punctuation of the manuscript need to be revised and improved throughout the text, and the manuscript should be revised by a native English speaker.

.

Reviewer #1: No

Reviewer #2: No

Reviewer #3: No

---

## [Author Response · Author response to Decision Letter 1]

6 Feb 2026

Point-by-point response to editors and reviewers

Authors’ responses to Editor

Manuscript ID：PONE-D-25-59759

Title：Association between Psychological Distress, Lifestyle and Career Planning on Health-Related Quality of Life among the University Students during School Closure of COVID-19 Pandemic in China

Authors：Baochen Su, Zhengna Sun, Rui Chen, Hui Liu, Xixing Xu, Fanlei Kong

The authors appreciate the editor and reviewers’ kind support on reviewing our manuscript and providing us with insightful and detailed comments. We highly value the reviewers’ thoughtful feedback and the important issues highlighted by them. We have addressed each reviewer’s comments in a point-by-point response below. For clarity, we have reproduced the reviewers’ original text in full, with “Q” representing their comments and “A” representing our responses (in red text). Moreover, the authors have highlighted all the modified parts in yellow so that they could be read more clearly by the editor and reviewers.

The authors carefully read the PLOS One guidelines for authors and performed a standardized conversion of the article structure, citations, graphics, and table formats, and invited a native English-speaking editor to grammatically assess this article to ensure compliance with the journal's publication standards.

Below are our point-to-point responses to reviewers' comments:

Authors’ responses to Reviewer 1

Q1：The study presents the results of original research

The submitted manuscript is a research article reporting findings from a cross-sectional study conducted among university students in China during the COVID-19 school closure period. The authors claim that this work is the first study to explore the determinants of health-related quality of life (HRQOL) among this population from the combined perspective of psychological distress, lifestyle habits, and career planning. The purpose of the study, which is to explore these associations, is clearly stated.

A1: Thank you for recognizing the originality and purpose of our study. We extend our heartfelt thanks to your thorough and fundamentally supportive review. Your endorsement of the study’s core contribution, coupled with their precise guidance for improvement, has made the revision process both rewarding and productive. We believe our research will be substantially stronger.

Q2: Results reported have not been published elsewhere

The submission is presented as a manuscript draft undergoing the peer review process. Based on the contents, which detail the methodology, novel statistical analyses, and conclusions derived from the collected data of 1965 respondents, the findings appear to be reported here for the first time. However, the authors do not state this explicitly.

A2: We are deeply appreciative of the reviewer’s positive remarks about the Originality of the Article. We confirm that the findings presented in this manuscript have not been published elsewhere and are not under consideration for publication in any other journal. We have added a statement to this effect in the cover letter and will include it in the “conflict of Interest” section of the manuscript for clarity.

The corresponding changes could be found on Line 493 – Line 496 in the revised manuscript as follow.

The authors declare that the research was conducted in the absence of any commercial or financial relationships that could be construed as a potential conflict of interest. We confirm that the findings presented in this manuscript have not been published elsewhere and are not under consideration for publication in any other journal.

Q3: Experiments statistics and analyses are performed to a high standard and are described in sufficient detail

The study utilized standard measures, including the SF-12 for HRQOL and the Kessler Psychological disease scale (K10) for psychological distress. The analyses employed Chi-squared tests and binary logistic regression models (Models 1, 2, and 3) to explore associations between variables and HRQOL components (PCS and MCS). The authors performed collinearity diagnostics and confirmed the absence of multicollinearity among independent variables across the models. However, the methodology section notes that due to the lack of a systematic scale, the measurement of lifestyle habits and career planning relied only on selected relevant indicators, which is a noted limitation. Furthermore, the non-random snowball sampling method employed may introduce bias in the sample representativeness.

A3: We appreciate the reviewer’s acknowledgment of our methodological approach. We have provided a clear description of these limitations in the limitations section of the article, and we will strive to address these issues in future research designs to achieve better results.

Q4: Conclusions are supported by the data and presented appropriately

The conclusions confirm that low psychological distress, a normal lifestyle, and clearer career planning were associated with higher HRQOL among the university students during the school closure. The presented logistic regression data supports this, showing strong odds ratios linking lower psychological distress to better PCS (OR=3.909) and MCS (OR=8.330) outcomes. Similarly, high physical activity intensity was positively associated with good PCS (OR=4.589). While the data supports the associations presented, the authors correctly identify a limitation inherent to the study design: the cross-sectional nature prevents the prediction of causal relationships.

A4: We sincerely thank the reviewer for their generous appraisal, particularly regarding our section of conclusion. Such recognition motivates us greatly, and we have endeavored to maintain these strengths for longitudinal studies in future research.

Q5: The article is intelligible and written in standard English

The article is generally intelligible and uses appropriate academic language throughout the Abstract, Methods, Results, and Discussion sections. The authors acknowledge that English language proofreading was conducted during the revision process.

A5: Thank you for your positive feedback regarding the language and readability of our manuscript. We are pleased that the language quality was found acceptable. We acknowledge that English language proofreading was indeed conducted during the revision process to ensure the quality and clarity of the manuscript. We appreciate your recognition of our efforts in this regard.

Q6: The research meets ethical standards and is ethically sound

The study meets ethical standards as confirmed by the ethics statement. The research design aligned with the tenets of the Declaration of Helsinki, 1996, and received formal approval from the Shandong University Institutional Ethics Committee (Task no. LL20220425). Written informed consent for participation was obtained from the participants’ legal guardians.

A6: Thank you very much for your recognition of the ethical aspects of this research. We confirm that the study was conducted in accordance with the Declaration of Helsinki and approved by the Shandong University Institutional Ethics Committee (Task no. LL20220425). We have explicitly restated this in the Ethics Statement section.

Q7: The article adheres to reporting guidelines and community data availability standards

The authors have provided necessary declarations regarding funding and competing interests, noting support from the National Natural Science Foundation of China (No. 71804094) and Fundamental Research Funds for the Central Universities (No. 2022KJGL01), and confirming that no competing interests exist. However, the authors state that adherence to community data availability standards is restricted. The data cannot be shared publicly because of concerns involving participant privacy.

A7: We appreciate the comment regarding data availability. We have clarified in the Data Availability Statement that due to ethical and privacy restrictions, the data cannot be shared publicly. However, we confirm that anonymized data is available upon reasonable request from the corresponding author, in compliance with institutional and ethical guidelines.

Once again, thanks very much for all your valuable and insightful comments.

Best wishes.

Fanlei Kong

Corresponding Author

Authors’ responses to Reviewer 2

Q1：Under the Materials and Methods, there should be exclusion and inclusion criteria to understand the category of group that the study considered. it is also reported that the snowballing sampling method was used in collecting the online data. It should be clearly stated in the manuscript how it was done during the data collection.

A1: Thank you for your insightful review of the manuscript and for your helpful comments on this section. This suggestion has been of great help in improving the quality of the manuscript, and the authors attach great importance to it. The authors have supplemented the details of the inclusion and exclusion criteria as well as the sample collection process, with the following specific changes. The revised manuscript is as followed.

The corresponding changes could be found on Line 195 – Line 221 in the revised manuscript as follow.

2.1. Data and Sample

Cross-sectional survey design and anonymous online questionnaire were used to assess the quality of life of university students during the school closure of COVID-19. The survey was conducted through an online platform called “Wenjuan.com” (commonly used in China), using a snowball sampling method. Initial respondents were recruited via university student WeChat groups; they were then encouraged to share the questionnaire link with their peers who also met the eligibility criteria, thereby expanding the sample through chain-referral. Data collection took place from May 5th to May 12th, 2022, during the period of school closure. All participants provided informed consent electronically by agreeing to participate before starting the questionnaire.

The snowball sampling method was used due to the difficulty of accessing the nationally dispersed university student population through traditional random sampling during strict campus lockdowns. This approach effectively reached the target population through participants' social networks in the absence of a reliable sampling frame.

Participants were recruited according to the following protocol-defined eligibility requirements: only Chinese students who were officially registered at a mainland university and were obliged to reside on campus under the institution’s COVID-19 “closed-campus” policy were eligible; international students, those living off-campus, and any individuals with a pre-existing major physical or psychiatric disorder that might independently affect the quality of life were excluded.

The minimum sample size was calculated using the cross-sectional formula n = [Z²₁₋α/₂ × P × (1 – P) / d², with Z = 1.96 (95% CI), P = 0.25, d = 0.05 and a design effect of 1.3, yielding 416 participants after 10% invalid allowance. To ensure statistical power and account for invalid responses, after excluding incomplete responses and malicious filling, a total of 1965 questionnaires were used in this study well exceeding the minimum requirement.

Q2: Under results, participants' characteristics are incomplete because we are sure of the age group of students you considered in the study and whether they are all nationals or not.

A2: Thank you for this helpful comment. The authors highly value your feedback and have made systematic responses and revisions. In response to your concern regarding the completeness of the participants’ characteristics, we have improved the participants' characteristics and revised Table 1 and the results.

1.Age: The overall mean age is 22.3 ± 1.7 years. Because the entire sample consisted of college students enrolled in mainland Chinese universities, this small dispersion reflects normal variation and has no clinical or statistical relevance for the present analyses; therefore, no additional categorization was applied.

2.Nationality: All participants were Chinese nationals holding mainland PRC identification. The following sentence has been added to the Participants subsection: “All students were Chinese studied in Chinese universities.”

3.Region: Table 1 now includes a “Region” variable, showing that 74.9% of participants were recruited from universities in northern China and 25.1% from southern China. Chi-square tests indicated no significant between-region differences for either PCS or MCS (P = 0.679 and 0.134, respectively), suggesting that regional distribution did not materially influence the main outcomes.

We believe these additions adequately clarify the homogeneity and representativeness of the sample and address the issue of incomplete participant characteristics.

The corresponding changes could be found on Line 275 – Line 292 in the revised manuscript as follow.

3.1. Basic characteristics of the participants

The basic information about the participants was provided in Table 1. All students were Chinese studied in Chinese universities with an average age of 22.25 ± 1.71. It was shown that 59.8% of the sample was female. 55.8% of participants were rural registered residence. As for the region of the participants, 74.9% of participants were recruited from universities in northern China and 25.1% from southern China. Chi-square tests indicated no significant between-region differences for either PCS or MCS (P = 0.679 and 0.134)

Table 1 Basic characteristics of the participants

Variables Total Good PCS Poor PCS P Good MCS Poor MCS P

n (%) n (%) n (%) n (%) n (%)

Age 22.25 ± 1.71 23.0±1.41 21.5±2.12 0.801 21.8±2.40 21.6±2.45 0.081

Gender <0.001 0.316

Male 790(40.2) 555(70.3) 235(29.7) 606(76.7) 184(23.3)

Female 1175(59.8) 943(80.3) 232(19.7) 898(75.1) 297(24.9)

hukou 0.022 0.305

Rural 1096(55.8) 814(74.3) 282(25.7) 818(74.6) 278(25.4)

City 869(44.2) 684(78.7) 185(21.3) 666(76.6) 203(23.4)

Region 0.679 0.134

northern China 1472(74,9) 1119(74.7) 353(75.6) 1100(74.1) 372(77.3)

southern China 493(25.1) 379(25.3) 114(24.4) 384(25.9) 109(22.7)

Once again, thanks very much for all your valuable and insightful comments.

Best wishes.

Fanlei Kong

Corresponding Author

Authors’ responses to Reviewer 3

Q1: The abstract should present a clear overview of the study's objectives and the general findings. However, the first paragraph should start with a sentence about the importance of the study and then the purpose of the research.

A1: We are very honored to receive your careful review and thank you for your constructive comments. Recognizing the importance of these points, we have undertaken a significant revision of the affected section. We have restructured the abstract as recommended. The first sentence now states the importance of the study, which is directly followed by the purpose of the research.

The corresponding changes could be found on Line 52 – Line 92 in the revised manuscript as follow.

Abstract: The COVID-19 pandemic and associated school closures presented unprecedented challenges to university students' well-being, highlighting an urgent need to understand the factors influencing their health-related quality of life. This study aimed to explore the association between psychological distress, lifestyle, and career planning on the health-related quality of life of university students during the school closure period in China. A cross-sectional study was conducted, collecting data from 1965 Chinese college students locked down in campus during COVID-19 using a snowball sampling method via an online questionnaire platform (Wenjuan.com). Psychological distress and the health-related quality of life were measured by scales. Descriptive analysis, Chi-squared test and logistic regression analysis were employed to analyze the data. The mean physical component summary score was 47.5, while the mean mental component summary score was 36.8. Students with low psychological distress, high physical activity intensity, a regular studying schedule arrangement, clear career planning during COVID-19 were more likely to report a good physic

---

## [Decision Letter · Decision Letter 1]

23 Feb 2026

Dear Dr. Kong,

Thank you for submitting your manuscript to PLOS ONE. After careful consideration, we feel that it has merit but does not fully meet PLOS ONE’s publication criteria as it currently stands. Therefore, we invite you to submit a revised version of the manuscript that addresses the points raised during the review process.

**ACADEMIC EDITOR:**

Dear Authors,

The manuscript has been reviewed by three reviewers. Only a few minor corrections remain.

We are asking you to make minor revisions before the paper can be published.

Specifically:

1. Abstract: You need to include the level or strength of the association (e.g., statistical values, strength of relationships, coefficients, etc.) between psychological distress, lifestyle, career planning, and health-related quality of life. Right now, the abstract mentions the association but does not indicate how strong it is.

2. Conclusion: The conclusion is considered confusing and should more clearly reflect the original aim of the study. It needs to align directly with what the research set out to investigate.

Overall, we find find the manuscript insightful and suitable for publication, pending these minor corrections.

We look forward to receiving your revised manuscript.

Kind regards,

Javier Fagundo-Rivera, PhD

Academic Editor

PLOS One

Journal Requirements:

**Additional Editor Comments:**

Dear Authors,

The manuscript has been reviewed by three reviewers. Only a few minor corrections remain.

We are asking you to make minor revisions before the paper can be published.

Specifically:

1. Abstract: You need to include the level or strength of the association (e.g., statistical values, strength of relationships, coefficients, etc.) between psychological distress, lifestyle, career planning, and health-related quality of life. Right now, the abstract mentions the association but does not indicate how strong it is.

2. Conclusion: The conclusion is considered confusing and should more clearly reflect the original aim of the study. It needs to align directly with what the research set out to investigate.

Overall, we find find the manuscript insightful and suitable for publication, pending these minor corrections.

Reviewers' comments:

Reviewer's Responses to Questions

**Comments to the Author**

Reviewer #1: All comments have been addressed

Reviewer #2: All comments have been addressed

Reviewer #3: All comments have been addressed

2. Is the manuscript technically sound, and do the data support the conclusions?

Reviewer #1: Yes

Reviewer #2: Yes

Reviewer #3: Yes

3. Has the statistical analysis been performed appropriately and rigorously?

Reviewer #1: Yes

Reviewer #2: Yes

Reviewer #3: Yes

4. Have the authors made all data underlying the findings in their manuscript fully available?

Reviewer #1: No

Reviewer #2: Yes

Reviewer #3: Yes

5. Is the manuscript presented in an intelligible fashion and written in standard English?

Reviewer #1: Yes

Reviewer #2: Yes

Reviewer #3: Yes

**Reviewer #1:**The manuscript has been re-evaluated exclusively with respect to the comments raised in my previous review and considering the authors’ revised submission and response letter. All issues raised in my previous review have been satisfactorily addressed in the revised manuscript.The manuscript has been re-evaluated exclusively with respect to the comments raised in my previous review and considering the authors’ revised submission and response letter. All issues raised in my previous review have been satisfactorily addressed in the revised manuscript.The manuscript has been re-evaluated exclusively with respect to the comments raised in my previous review and considering the authors’ revised submission and response letter. All issues raised in my previous review have been satisfactorily addressed in the revised manuscript.The manuscript has been re-evaluated exclusively with respect to the comments raised in my previous review and considering the authors’ revised submission and response letter. All issues raised in my previous review have been satisfactorily addressed in the revised manuscript.

**Reviewer #2:**I have read through the manuscript and found it insightful. I therefore recommend this paper to be published with minor corrections based on the following:I have read through the manuscript and found it insightful. I therefore recommend this paper to be published with minor corrections based on the following:I have read through the manuscript and found it insightful. I therefore recommend this paper to be published with minor corrections based on the following:I have read through the manuscript and found it insightful. I therefore recommend this paper to be published with minor corrections based on the following:

The abstract summarises the work, giving the association between psychological distress, lifestyle and career planning on the health-related QoL of university students, but failed to include the level of association in the abstract.

The main paper introduces COVID-19, the students and their demographic factors, as well as the health-related quality of life. They further show an account of the reviewed literature. The methods are also well explained. The results and discussion were also well written, but the conclusion was confusing. It should reflect what they aimed to do.

**Reviewer #3:**(No Response)(No Response)(No Response)(No Response)

.

Reviewer #1: No

Reviewer #2: No

Reviewer #3: No

---

## [Author Response · Author response to Decision Letter 2]

4 Mar 2026

Point-by-point response to editors and reviewers

Authors’ responses to Editor

Manuscript ID：PONE-D-25-59759

Title：Association between Psychological Distress, Lifestyle and Career Planning on Health-Related Quality of Life among the University Students during School Closure of COVID-19 Pandemic in China

Authors：Baochen Su, Zhengna Sun, Rui Chen, Hui Liu, Xixing Xu, Fanlei Kong

The authors appreciate the editor and reviewers’ kind support on reviewing our manuscript and providing us with insightful and detailed comments. We highly value the editor and reviewers’ thoughtful feedback and the important issues highlighted by them. We have addressed the editor and each reviewers’ comments in a point-by-point response below. For clarity, we have reproduced the reviewers’ original text in full, with “Q” representing their comments and “A” representing our responses (in red text). Moreover, the authors have highlighted all the modified parts in yellow for the better reading experience by the editor and reviewers.

Below are our point-to-point responses to reviewers' comments:

Authors’ responses to Academic Author

Q1: Abstract: You need to include the level or strength of the association (e.g., statistical values, strength of relationships, coefficients, etc.) between psychological distress, lifestyle, career planning, and health-related quality of life. Right now, the abstract mentions the association but does not indicate how strong it is.

A1: Thank you for your insightful and helpful comments on including the level/strength of the association in the ABSTRACT. This suggestion has been of great help in improving the quality of the manuscript, and the authors attach great importance to it. We have added the level/strength of the association between psychological distress, lifestyle, career planning, and health-related quality of life in the abstract. The revised manuscript is as follow.

Revised Manuscript (Line 31 – Line 54):

Abstract: The COVID-19 pandemic and associated school closures presented unprecedented challenges to university students' well-being, highlighting an urgent need to understand the factors influencing their health-related quality of life. This study aimed to explore the association between psychological distress, lifestyle, and career planning on the health-related quality of life of university students during the school closure period in China. A cross-sectional study was conducted, collecting data from 1965 Chinese college students locked down in campus during COVID-19 using a snowball sampling method via an online questionnaire platform (Wenjuan.com). Psychological distress and the health-related quality of life were measured by scales. Descriptive analysis, Chi-squared test and logistic regression analysis were employed to analyze the data. The mean physical component summary score was 47.5, while the mean mental component summary score was 36.8. Students who were women (OR=1.444, P=0.003), having a boy or girl friend (OR=1.379, P=0.008), with rural hukou (OR=1.446, P=0.004), with low psychological distress (OR=4.589, P<0.001), high physical activity intensity (OR=3.909, P<0.001), a regular studying schedule arrangement (OR=2.553, P=0.008), clear career planning (OR=1.570, P=0.001) during COVID-19 were more likely to report a good physical component summary. For mental component summary, lower psychological distress (OR=8.330, P<0.001), a regular studying schedule arrangement (OR=2.892, P=0.001) and keeping same job-hunting pressure (OR=1.852, P=0.003) were positive influencing factors, whereas having a boy or girl friend (OR=0.774, P=0.032) and having no clear career planning (OR=0.752, P=0.020) during COVID-19 were negative influencing factors. In conclusion, while health-related quality of life was generally good, lower psychological distress, healthier lifestyle habits, and clearer career planning were significantly associated with better health-related quality of life among university students during the pandemic-related school closure. These findings underscore the importance of integrating mental health support, lifestyle guidance, and career planning services into student support systems during public health crises. Future studies should develop and evaluate interventions targeting these modifiable factors.

Q2: Conclusion: The conclusion is considered confusing and should more clearly reflect the original aim of the study. It needs to align directly with what the research set out to investigate.

A2: The authors greatly appreciate the editor’s constructive and valuable comments on the CONCLUSION. Following the editor’s important suggestion, we have carefully revised the conclusion to make it clearer. The revised conclusion now directly aligns with the original aim of the study and explicitly reflects the key issues that this research intended to investigate. The revised manuscript is as followed.

Revised Manuscript (Line 349 – Line 365):

In summary, to the best of our knowledge, this was the first study to explore the determinants of HRQOL among Chinese university students during the school closure of the COVID-19 from the perspective of psychological distress, lifestyle habits and life planning. Lower psychological distress, better lifestyle and clearer career planning were significantly associated with the better HRQOL of university students during the school closure of the COVID-19. Maintaining a regular studying schedule arrangement and a relatively high-intensity of exercise during the school closure of the COVID-19 may have a good impact on maintaining a good HRQOL of university students. Intervention measures derived from this research could be put into practice for governments and universities in the future public health emergencies.

Once again, thanks very much for all your valuable and insightful comments.

Best wishes.

Fanlei Kong

Corresponding Author

Authors’ responses to Reviewer 1

Q1: The manuscript has been re-evaluated exclusively with respect to the comments raised in my previous review and considering the authors’ revised submission and response letter. All issues raised in my previous review have been satisfactorily addressed in the revised manuscript.

A1: Thank you very much for the valuable comments you provided during the review process. Your feedback has played a significant role in improving the quality of this paper. Once again, thank you for your help and support.

Once again, thanks very much for all your valuable and insightful comments.

Best wishes.

Fanlei Kong

Corresponding Author

Authors’ responses to Reviewer 2

Q1: The abstract summarises the work, giving the association between psychological distress, lifestyle and career planning on the health-related QoL of university students, but failed to include the level of association in the abstract.

A1: The authors are very honored to receive your insightful review and thank you for your constructive and valuable comments. Recognizing the importance of these points, we have added the coefficient of the association between psychological distress, lifestyle, career planning, and health-related quality of life in the abstract. The revised manuscript is as followed.

Revised Manuscript (Line 31 – Line 54):

Abstract: The COVID-19 pandemic and associated school closures presented unprecedented challenges to university students' well-being, highlighting an urgent need to understand the factors influencing their health-related quality of life. This study aimed to explore the association between psychological distress, lifestyle, and career planning on the health-related quality of life of university students during the school closure period in China. A cross-sectional study was conducted, collecting data from 1965 Chinese college students locked down in campus during COVID-19 using a snowball sampling method via an online questionnaire platform (Wenjuan.com). Psychological distress and the health-related quality of life were measured by scales. Descriptive analysis, Chi-squared test and logistic regression analysis were employed to analyze the data. The mean physical component summary score was 47.5, while the mean mental component summary score was 36.8. Students who were women (OR=1.444, P=0.003), having a boy or girl friend (OR=1.379, P=0.008), with rural hukou (OR=1.446, P=0.004), with low psychological distress (OR=4.589, P<0.001), high physical activity intensity (OR=3.909, P<0.001), a regular studying schedule arrangement (OR=2.553, P=0.008), clear career planning (OR=1.570, P=0.001) during COVID-19 were more likely to report a good physical component summary. For mental component summary, lower psychological distress (OR=8.330, P<0.001), a regular studying schedule arrangement (OR=2.892, P=0.001) and keeping same job-hunting pressure (OR=1.852, P=0.003) were positive influencing factors, whereas having a boy or girl friend (OR=0.774, P=0.032) and having no clear career planning (OR=0.752, P=0.020) during COVID-19 were negative influencing factors. In conclusion, while health-related quality of life was generally good, lower psychological distress, healthier lifestyle habits, and clearer career planning were significantly associated with better health-related quality of life among university students during the pandemic-related school closure. These findings underscore the importance of integrating mental health support, lifestyle guidance, and career planning services into student support systems during public health crises. Future studies should develop and evaluate interventions targeting these modifiable factors.

Q2: The main paper introduces COVID-19, the students and their demographic factors, as well as the health-related quality of life. They further show an account of the reviewed literature. The methods are also well explained. The results and discussion were also well written, but the conclusion was confusing. It should reflect what they aimed to do.

A2: The authors sincerely appreciate the reviewer for the positive feedback and constructive comments on our manuscript. We are grateful that the introduction, literature review, methodology, results, and discussion were recognized as clear and well presented.

Following the reviewer’s valuable suggestion, the authors have thoroughly revised the CONCLUSION section. The revised parts now directly corresponds to the original aims of the study and clearly summarizes the key findings in line with what the research intended to investigate. The revised manuscript is as followed.

Revised Manuscript (Line 349 – Line 365):

In summary, to the best of our knowledge, this was the first study to explore the determinants of HRQOL among Chinese university students during the school closure of the COVID-19 from the perspective of psychological distress, lifestyle habits and life planning. Lower psychological distress, better lifestyle and clearer career planning were significantly associated with the better HRQOL of university students during the school closure of the COVID-19. Maintaining a regular studying schedule arrangement and a relatively high-intensity of exercise during the school closure of the COVID-19 may have a good impact on maintaining a good HRQOL of university students. Intervention measures derived from this research could be put into practice for governments and universities in the future public health emergencies.

Once again, thanks very much for all your valuable and insightful comments.

Best wishes.

Fanlei Kong

Corresponding Author

---

## [Editor Report · Decision Letter 2]

26 Mar 2026

Association between Psychological Distress, Lifestyle and Career Planning on Health-Related Quality of Life among the University Students during School Closure of COVID-19 Pandemic in China

PONE-D-25-59759R2

Dear Dr. Kong,

We’re pleased to inform you that your manuscript has been judged scientifically suitable for publication and will be formally accepted for publication once it meets all outstanding technical requirements.

Kind regards,

Javier Fagundo-Rivera, PhD

Academic Editor

PLOS One

**Additional Editor Comments:**

Dear Authors,

Following this second round of revisions, we are pleased to inform you that both the reviewers and the editor are fully satisfied with the changes implemented in your manuscript. The article is now ready for publication.

We would like to sincerely thank you for your effort, dedication, and the care you have taken in addressing all comments. We also appreciate the trust you have placed in the journal by choosing it as the venue for your work.

Congratulations on this achievement.

Kind regards.
---

## [Editor Report · Acceptance letter]

PONE-D-25-59759R2

PLOS One

Dear Dr. Kong,

I'm pleased to inform you that your manuscript has been deemed suitable for publication in PLOS One. Congratulations! Your manuscript is now being handed over to our production team.

Kind regards,

on behalf of

Dr. Javier Fagundo-Rivera

Academic Editor

PLOS One